# A persistent variant telomere sequence in a human pedigree

Angela M. Hinchie [1,2], Samantha L. Sanford[3,4], Kelly E. Loughridge[1,2], Rachel M. Sutton[1,2], Anishka H. Parikh[1,2], Agustin A. Gil Silva[1,2], Daniel I. Sullivan[1,2], Pattra Chun-On[1,2], Matthew R. Morrell[2], John F. McDyer[2], Patricia L. Opresko [3,4,5] & Jonathan K. Alder [1,2] ✉

The telomere sequence, TTAGGG, is conserved across all vertebrates and plays an essential role in suppressing the DNA damage response by binding a set of proteins termed shelterin. Changes in the telomere sequence impair shelterin binding, initiate a DNA damage response, and are toxic to cells. Here we identify a family with a variant in the telomere template sequence of telomerase, the enzyme responsible for telomere elongation, that led to a non-canonical telomere sequence. The variant is inherited across at least one generation and one family member reports no significant medical concerns despite ~9% of their telomeres converting to the novel sequence. The variant template disrupts telomerase repeat addition processivity and decreased the binding of the telomere-binding protein POT1. Despite these disruptions, the sequence is readily incorporated into cellular chromosomes. Incorporation of a variant sequence prevents POT1-mediated inhibition of telomerase suggesting that incorporation of a variant sequence may influence telomere addition. These findings demonstrate that telomeres can tolerate substantial degeneracy while remaining functional and provide insights as to how incorporation of a non-canonical telomere sequence might alter telomere length dynamics.

The telomere sequence is ancient. TTAGGG, the vertebrate telomeric sequence, has been conserved for at least 400 million years[1], and is restricted by the sequence-specific binding of shelterin[2,3]. The shelterin complex is composed of six proteins (TRF1, TRF2, RAP1, TIN2, POT1, and TPP1) that bind both the double- and single-stranded portions of the telomere in a sequence-specific manner and function to suppress the DNA damage response (DDR) at free ends of linear chromosomes[4]. The POT1-TPP1 heterodimer binds the single-stranded G-rich overhang at 3' end of the chromosome and contributes to telomere length regulation[3,5,6]. Loss of shelterin leads to a DDR, chromosome fusion, and genome instability[4,7].

As cells divide, the ends of chromosomes are not completely replicated, leading to erosion of the telomere over time. Telomerase compensates for DNA loss by adding telomere repeats to chromosome ends. Telomerase is recruited to the telomere by POT1-TPP1 which also facilitates repeat addition processivity (RAP) of the enzyme[5,8]. Paradoxically, while important for telomerase RAP, POT1 binding to the end of the telomere prevents telomerase access and inhibits extension[6,9]. During elongation, telomerase associates with the 3' end of existing telomeres using the alignment region of the non-coding telomerase RNA (TR; encoded by *TERC*) to reverse transcribe new telomeric sequence[10]. While the non-template portion of the TR

[1]Dorothy P. and Richard P. Simmons Center for Interstitial Lung Disease, University of Pittsburgh, Pittsburgh, PA, USA. [2]Division of Pulmonary, Allergy, Critical Care, and Sleep Medicine, University of Pittsburgh, Pittsburgh, PA, USA. [3]Environmental and Occupational Health Department, School of Public Health, University of Pittsburgh, Pittsburgh, PA, USA. [4]University of Pittsburgh Medical Center, Hillman Cancer Center, Pittsburgh, PA, USA. [5]Pharmacology and Chemical Biology Department, University of Pittsburgh, Pittsburgh, PA, USA. ✉e-mail: jalder@pitt.edu

sequence is divergent between species, the template is tightly conserved owing to its function in determining the telomere sequence[11]. Varying the terminal telomere sequence by overexpression of TR with a variant template region elicits a DDR that inhibits cell proliferation[12–16].

Defects in telomere maintenance cause a syndrome in humans characterized by age-dependent clinical phenotypes and genetic anticipation[17–19]. Phenotypes in these patients are driven by short telomeres and manifest when telomeres reach a functional threshold[17,20]. Telomerase is not expressed in the majority of somatic cells and in cells that do, its expression is tightly controlled, and even a small decrease in overall telomerase activity can eventually lead to clinical consequences via inheritance of shorter and shorter telomeres[21]. Critically short telomeres trigger a DDR that can lead to replicative senescence and manifest as a spectrum of clinical phenotypes[22–25]. Among the clinical phenotypes, idiopathic pulmonary fibrosis (IPF) is the most common, and is often associated with mutations in telomerase[19,26].

## Results

### A variant telomere sequence in a human pedigree

We recently characterized a large cohort of patients with IPF and identified a number of patients with rare variants in *TERT* and *TERC*[27]. One of the patients we examined carried a heterozygous C>A transversion (r.50C>A) within the template region of TR. The patient's medical history was notable for premature greying and the development of IPF at a young age (43 years), but was otherwise unremarkable (Fig. 1a,b). Analysis of the proband's son (17 years) indicated that he also carried the variant but reported no significant medical concerns. The variant, r.50C>A, is predicted to add a non-canonical telomere sequence, TTAGGT, rather than TTAGGG (Fig. 1c).

To determine if the r.50C>A variant-containing telomerase was active in vivo, we examined whole genome sequencing (WGS) data of the proband and his son to test if variant sequences were incorporated into telomeres. We estimated the composition of wild-type and variant repeats in the proband, son, and 10 controls by examining available WGS data from peripheral blood mononuclear cells (PBMCs) (proband and controls) or a saliva sample (son). We examined the raw sequencing files and arbitrarily defined reads as "telomeric" if they contained 13 repeats of the wild-type or variant repeats (>50% of the read length; Supplementary Fig. 1e). Telomere reads were trimmed for quality (see methods), and the number of wild-type and variant repeats were quantified using custom Perl scripts (Supplementary Fig. 1a–c). The proband's telomeres were composed of roughly 2.7% of the variant sequence (TTAGGT) compared to 0.4% in the control samples, a six-fold increase, while the proband's son's telomeres contained 9.2% variant sequence, a 23-fold increase (Fig. 1e), indicating that telomerase had added significant variant repeats in the germline or during development. The number of canonical repeats remained consistent in the proband (84.3%) but decreased in his son (77.8%) compared to controls (84.4%) (Supplementary Fig. 1d). Approximately 15% percent of the telomere was composed of sequence that was neither wild type nor TTAGGT (Supplementary Fig. 1d); similar to previous studies that analyzed telomere composition by WGS data[28–30]. Noncanonical telomeric sequences could arise from telomere degeneracy, sub-telomeric sequences, intrachromosomal telomeric sequences, or sequencing errors. Analysis of the repeat pattern showed consecutive groups of TTAGGT repeats in both father and son but not in controls, suggesting the increased variant sequence did not arise randomly or due to sequencing errors (Fig. 1d, Supplementary Fig 3a). Together, the available WGS data from the proband and his son support that the variant template sequence functions within telomerase and can be added and maintained within telomeres and inherited across generations.

We next investigated if the variant telomere sequences could be detected in situ. We designed peptide nucleic acid probes that were specific for three consecutive variant or wild-type repeats and

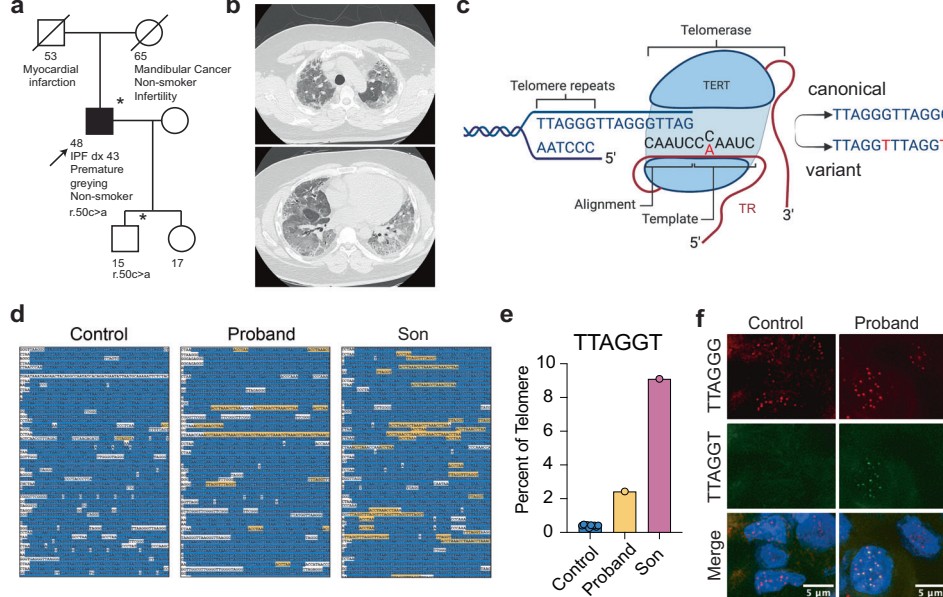

**Fig. 1 | A variant in telomerase RNA template alters the canonical telomere sequence. a** Pedigree of the family carrying the *TERC* r.50 C > A variant. The proband (arrow) was diagnosed with IPF at age 43. Asterisks indicate individuals from which DNA was available. Squares indicate males and circles indicate females. A line through the symbol indicates the individual is deceased. **b** Apical (upper) and basal (lower) chest CT showing interstitial changes and advanced fibrosis in the proband before lung transplantation. **c** Graphic showing the location of the patient-derived variant in the template of TR and the resulting telomere sequence. Figure was created with BioRender.com and released under a Creative Commons Attribution-NonCommercial-NoDerivs 4.0 international license. **d** Excerpts of telomeric sequences from whole genome sequencing of an individual with no template mutation, the proband, and the proband's son. Canonical repeats are highlighted in blue, variant TTAGGT repeats are highlighted in yellow, additional non-canonical sequences are in white. **e** The percentage of TTAGGT repeats in controls, the proband, and the proband's son. **f** PNA-FISH for wild-type (red) and variant (green) sequence in a tissue section of the proband's explanted lung and a control donated lung. Source data for (e) is provided as a Source Data file.

performed fluorescent in situ hybridization (FISH). The proband had undergone a lung transplant during his clinical care and we obtained explanted lung samples and probed for the wild-type and variant telomeres using FISH to the respective sequences. The variant telomere sequence was detected in the explanted lung samples but not in control donated lungs (Fig. 1f). The sequencing data and in situ hybridization strongly support that the variant sequences are incorporated by telomerase and persist within adult tissues.

## Sequence-mediated altered telomerase activity

We next sought to characterize the biochemical function of telomerase with a variant RNA template. During the telomerase catalytic cycle, telomerase adds six nucleotides and then either dissociates from the telomere or translocates to realign the template to add another six base pairs—termed repeat addition processivity (RAP). Based on the position of the variant within the template, we hypothesized the addition of the variant TTAGGT sequence to the end of the telomere would lead to a mismatch in the alignment of TR at r.C56 with the 3′ end of the variant telomere and cause a defect in RAP (Fig. 2a). To investigate this possibility, we generated expression constructs for the WT and C50A TR, as well as C50/56A TR, a double mutant that corrects the above-described mismatch by introducing a compensatory mutation in the alignment region of TR (Fig. 2b). We tested the TR variants in a direct telomerase activity assay with wild-type TERT (Fig. 2c). All TR constructs were able to extend from both a wild-type telomere primer (-GGTTAG) and the variant telomere primer (-GTTTAG). There was no decrease in overall telomerase activity with either C50A or C50/56A, as measured by the proportion of end-labeled primer that was extended during the reaction (Supplementary Fig. 2a). This suggests that the variant template does not disrupt the initial round of telomere addition. While the initial repeat addition was unaffected, RAP was significantly inhibited in reactions containing the C50A and C50/56A variants (Fig. 2c). The traditional method of calculating processivity did not differentiate between C50A and C50/56A despite an observable difference in the gel (Supplementary Fig. 2b). Therefore, we quantified the percentage of end-labeled primer extended beyond one repeat. This method revealed that the mismatch corrected C50/56A was 2.4-fold more likely to add two or more repeats than C50A, although still only half as likely as WT to do so (Fig. 2e). Mixing of the WT- and C50A-TR variants did not inhibit the activity of the WT, suggesting that C50A and the TTAGGT sequence do not interfere with the activity of the WT telomerase (Supplementary Fig. 2c). We tested if the addition of POT1-TPP1, a known enhancer of RAP, could rescue the low RAP in reactions with C50A. POT1-TPP1 was able to stimulate RAP in WT and C50/56A but not in C50A, further indicating that the mismatch at position 56 contributes to the loss in RAP (Fig. 2d, Supplementary Fig. 2d). Together, these data suggest that the variant telomerase is catalytically active but that the telomere-template mismatch causes a significant deficit in RAP.

We noted that both C50A and C50/56A showed a different pausing pattern than WT telomerase in assays of telomerase activity. Both variants paused one nucleotide early, opposite rU47 instead of rC46 (Fig. 2c) in our direct telomerase activity assays. Prior biochemical studies showed that telomerase pauses three nucleotides after the addition of the first T in the telomeric repeat, regardless of the location of the initial T[31]. Later structural studies have shown that telomerase utilizes a zipper head mechanism, with the template-primer duplex passing over a leucine "zipper" that prevents the formation of long heteroduplexes[32]. Structural modeling shows that a dG:rC pairing over this zipper is more energetically stable than a dT:rA pairing. The energetic difference in moving from a dG:rC to dT:rA pair has been predicted to cause telomerase to pause[32]. For C50A and C50/56A TR, the dG:rC to dT:rA transition occurs one nucleotide sooner, explaining the observed early pause and connecting the prior biochemistry and

structural experiments with the novel activity of this variant telomerase (Fig. 2f).

Telomerase activity may differ in vitro versus in vivo due to cellular factors that influence telomere elongation. To access the RAP of telomerase in vivo, we examined the WGS data to determine the frequency of single variant repeats versus multiple tandem variant repeats (Supplementary Fig. 3a, b). We first examined the expression of both alleles of TR in long non-coding RNA-sequencing (lncRNA-seq) and found that both alleles of TR were expressed in PBMCs from the proband ($n = 1$; Supplementary Fig. 3c) supporting that the variant TR had not been silenced. We noted in the whole genome sequencing data that the variant telomere repeats appeared in one or greater consecutive repeats, and the frequency of having multiple adjacent TTAGGT repeats decreased from 2 to 3 to 4 in a logarithmic fashion (Fig. 2g). We created a model describing the probability that the variant sequence exists in consecutive repeats and estimated the probability of extension based on sequencing data (Fig. 2h). Using this method to measure the processivity of both the variant and wild-type telomerase, we found the probability of processive addition by the variant TR in the proband and his son was 39% and 28% respectively, and by the wild-type TR in the proband and his son was 91% and 83%, respectively (Fig. 2i). Similar to the in vitro data, the in vivo RAP measurement shows a marked decrease in RAP for C50A TR.

## Addition of the variant telomere in vivo

We next examined the dynamics and consequences of in vivo addition of the variant sequence in TERT-positive cell lines. FISH with peptide nucleic acid (PNA) probes has previously been used to quantify nascent telomere addition in situ[16] and we adapted this method to monitor addition of the variant sequence identified in this study. We transduced cells with a lentivirus containing either an empty vector (EV), WT, C50A, or C50/56A TR. Variant telomere incorporation was measured by PNA-FISH probes complementary to the wild-type and variant sequences. In hTERT-RPE cells, 6 days post-transduction, 36.1% of C50A and 69.5% of C50/56A transduced cells had at least 10 variant telomere foci (Fig. 3a, b). Additionally, cells transduced with C50/56A had variant telomere intensities that were 4.8-fold brighter per cell than those transduced with C50A, demonstrating that the processivity deficit translates to less telomere addition in vivo (Fig. 3c). These results were recapitulated in LOX Melanoma and HCT116 + hTERT cells (Supplementary Fig. 4a–f). We next tested if differences in telomere addition remained consistent over a longer period of time. We chose LOX Melanoma cells for these studies as this line appears more tolerant of variant sequences (shown in Fig. 4). LOX Melanoma cells were transduced with wild-type and variant TR and metaphase spreads were analyzed at 9- and 85-days post-transduction. While C50/56A added significantly more than C50A TR at 9 days, by 85 days this difference was extinguished, suggesting that cells had reached a new telomere length homeostasis. By this timepoint the vast majority of wild-type telomere foci had an associated variant focus (Fig. 3f, g, Supplementary Fig. 4g–i).

While highly sensitive, measuring variant telomere addition by FISH cannot directly compare wild-type to variant telomere addition due to differences in hybridization of the distinct probes. We therefore used a Terminal Restriction Fragment (TRF) analysis to evaluate the extent of telomere lengthening in cells overexpressing the wild-type and variant sequences. We chose HCT116 cells as they have relatively short telomeres and are fairly tolerant of variant telomere addition[12]. We transduced cells with TERT and TERC constructs and collected genomic DNA for a TRF southern blot 15 days post-transduction. Cells transduced with the C50A variant lengthened telomeres to a much lesser extent than either C50/56A or WT, with C50/56A and WT reaching approximately similar lengths (Fig. 3d, e). Together, these data strongly support that the variant sequence can be added to

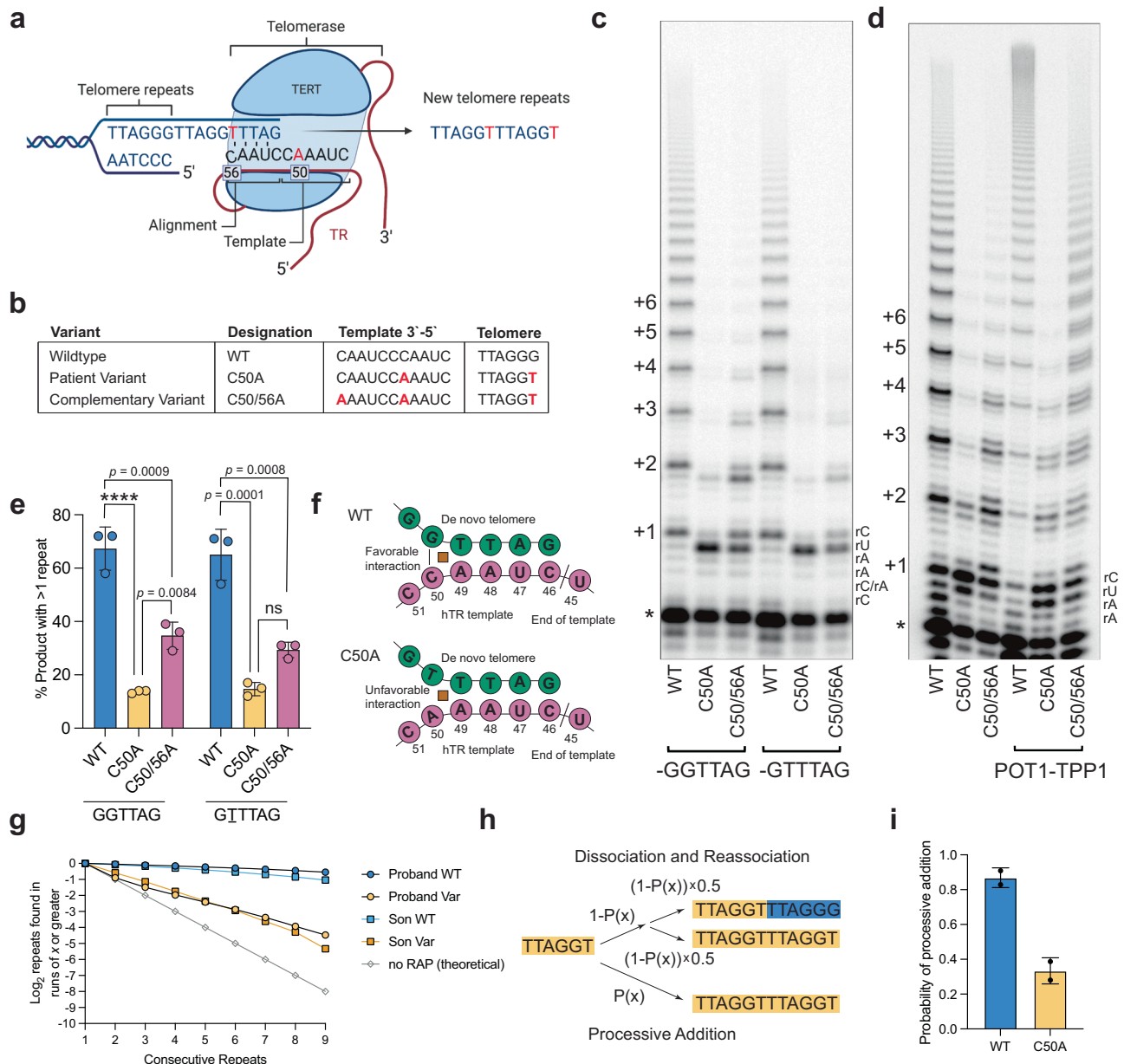

**Fig. 2 | The variant template compromises repeat addition processivity.**
**a** Schematic showing the template-telomere mismatch after the addition of the variant telomere sequence. Graphic was created with BioRender.com and released under a Creative Commons Attribution-NonCommercial-NoDerivs 4.0 international license. **b** Variant templates used and the resulting telomere sequence. Changes to the sequence are shown in red. **c** Telomerase direct assay with WT, C50A, and C50/56A telomerase RNA templates. RAP was measured extending from both a wild-type (-GGTTAGx3) and variant (-GTTTAGx3) primer. * Indicates the location of the radiolabeled primer and the numbers indicate the number of hexanucleotide repeats that were added. **d** Telomerase direct assay with primer A5, (TTAGGGTTAGCGTTAGGG) with the addition of POT1-TPP1 processivity factors. Primer A5 blocks 3` POT1 binding while allowing for processive addition[5]. * Indicates the location of the radiolabeled A5 primer and the numbers on the side of the

gel indicate the number of telomeric repeats that were added. The A5 primer has two additional Gs at the three prime end so that only 4 nucleotides are added in the first cycle. **e** Quantification of the percentage of product that extended beyond one repeat from (c). Mean±s.d is shown, *n* = 3 biological replicates, and groups were compared with one-way ANOVA with Tukey's multiple comparisons test. **f** Schematic showing the expected conformation of WT and C50A TR with the nascent telomere over the leucine zipperhead described in Wan et al.[32] **g** Proportion of TTAGGG (blue/light blue) and TTAGGT (yellow/orange) consecutive repeats found in WGS data from the proband and his son. Lines are also shown for theoretical numbers of consecutive repeats for non-processive addition (gray line). **h** Model of in vivo processivity. **i** Probability of processive addition from the WGS data, mean±s.d is shown, *n* = 2 (proband and son). ns, non-significant *p* ≥0.05, ****p* < 0.0001. Source data are provided as a Source Data file.

telomeres and that the r.56 mismatch significantly inhibits efficient addition in vivo.

### Cellular consequences of variant telomere sequences
We next examined the cellular consequences of variant sequence addition. Previous studies have shown that the incorporation of variant telomere sequences induced a DDR and impaired cell

proliferation[12,14,33]. The identification of a family that expressed a variant sequence and our whole genome analysis suggests that the TTAGGT sequence encoded by the patient's *TERC* variant may be at least partially tolerated. To examine the effect of the variant sequence on cell proliferation we over-expressed TRs encoding WT, C50A, C50/56A, and AU5 variants from lentiviral vectors co-expressing GFP in LOX Melanoma and hTERT-RPE cells (Supplementary Fig. 5a). The AU5

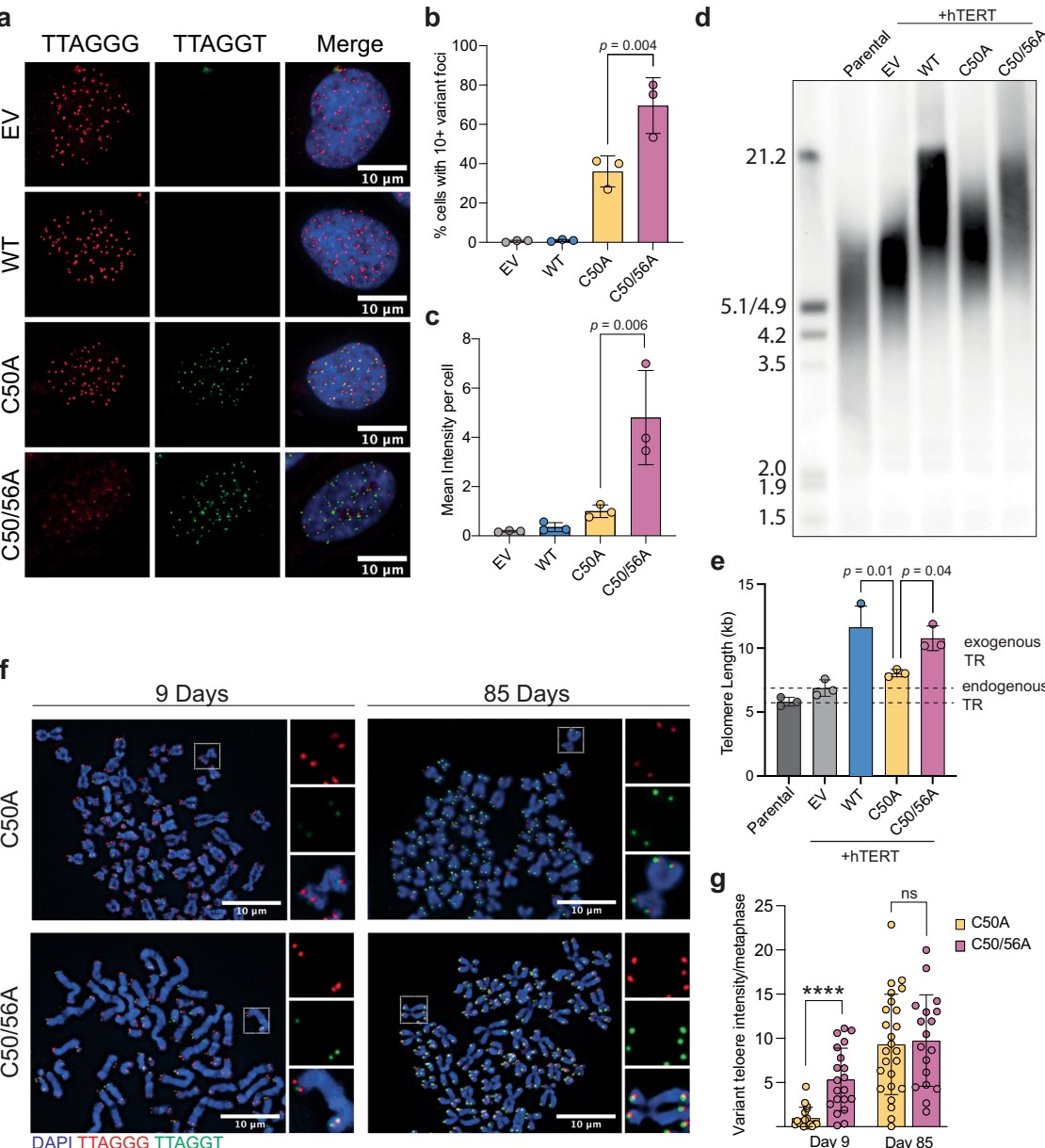

**Fig. 3 | Incorporation and persistence of a variant telomere sequence in cells.** **a** Representative photomicrograph of interphase FISH with PNA probes to the wild-type and variant sequence in hTERT-RPE cells 6 days post-transduction with empty vector (EV), WT, C50A or C50/56A TR. **b** Quantification of the percentage of cells with 10 or more visible variant telomere foci in (**a**). **c** Mean intensity per cell of the variant telomeres in (**a**) normalized to C50A. For (**b**) and (**c**), n = 3 biological replicates and mean±s.d is shown and at least 60 cells were evaluated in each independent experiment (see Source Data for exact numbers). **d** Telomere restriction fragment (TRF) southern blot of HCT116 cells co-transduced with hTERT and either and empty vector (EV), WT, C50A, or C50/56A TR, 15 days post-transduction. **e** Quantification of TRF southern blot in (**d**), n = 3 biological replicates, mean ± s.d. is shown. **f** Representative photomicrograph of metaphase FISH of LOX Melanoma cells with the wild-type and variant sequence at 9- and 85-days post-transduction with C50A or C50/56A TR. **g** Quantification of variant telomere intensity per metaphase of (**f**) normalized to C50A Day 9. Total metaphases analyzed as follows: C50A Day 9 (18), C50/56A Day 9 (22), C50A Day 85 (25), C50/56A Day 85 (20), median±s.d. is shown. Groups were compared with a two-tailed Student's unpaired *t* test. Groups in (**b**), (**c**), and (**e**) were compared with one-way ANOVA and Tukey's multiple comparison test. For all pair-wise comparisons, ns, non-significant $p \geq 0.05$, ****$p < 0.0001$. Source data are provided as a Source Data file.

variant encodes TATATA telomere repeats and has been previously reported to be toxic when expressed in cells[12,14,33].

We first examined cell proliferation by measuring the proportion of the surface area that was covered by transduced cells. In LOX Melanoma cells, only transduction with the AU5 TR template showed a significant decrease in proliferation whereas TTAGGT addition was tolerated. In contrast, in hTERT-RPE cells only the non-processive C50A variant template grew well while C50/56A moderately and AU5 significantly inhibited proliferation (Fig. 4a–c). To examine

growth more closely, we used a sensitive competition assay, in which approximately 50% of cells were transduced and the proportion of GFP-positive cells was monitored every 3–4 days to test whether expression of the variant TR influences the proliferation rate of transduced cells compared to non-transduced cells in the same well (Supplementary Fig. 5b, Supplementary Fig. 6). The results from the competition assay were similar to our previous results, with hTERT-RPEs being more sensitive to the variant telomere addition. In both LOX Melanoma and hTERT-RPE cells, the addition of the more

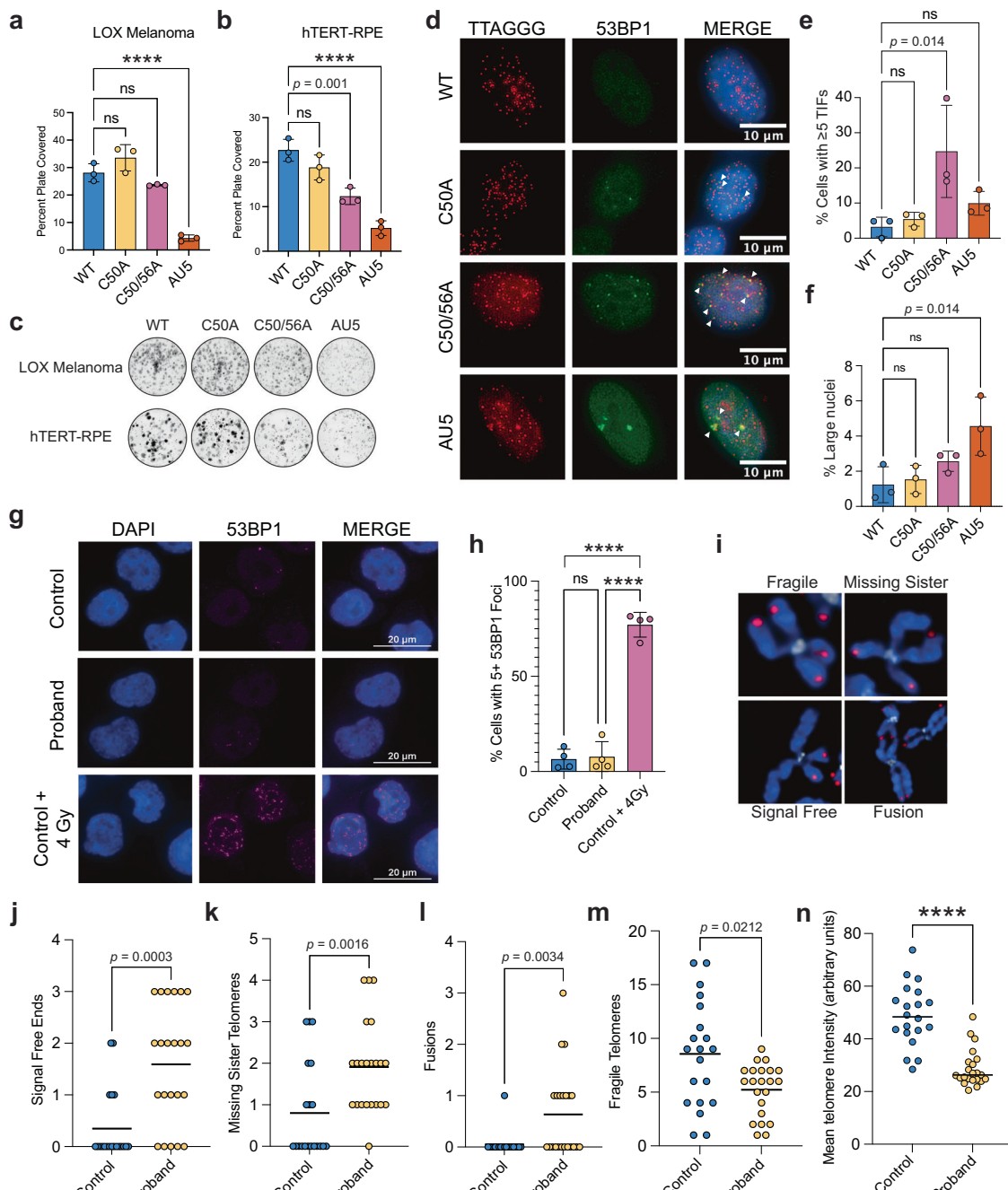

**Fig. 4 | Effects of variant sequence addition on DDR and chromosome stability.**
**a**, **b** Proliferation assays for LOX Melanoma and hTERT-RPE cells following transduction with lentiviruses expressing the indicated TR. Data are from $n = 3$ biological replicates. Mean±s.d. is shown. **c** Representative images of GFP-positive cells in the proliferation assay 7- (LOX Melanoma) or 9-days (hTERT-RPE) after transduction. **d** Representative photomicrograph of 53BP1 and telomere FISH staining in hTERT-RPE cells transduced with variant TR, 6 days post-transduction. **e** Quantification of telomere-induced foci (TIFs), of 53BP1 colocalized with the telomere of the cells in (**d**). **f** Quantification of large nuclei (>200 μm²) of the cells in (**d**). For (**e**) and (**f**) data are $n = 3$ biological replicates and mean±s.d. is shown. **g** Representative photomicrograph of 53BP1 staining in the proband and control lymphoblast, with a 4 Gy irradiated control. **h** Quantification of 53BP1 foci from (**g**), $n = 4$ independent experiments, mean±s.d is shown. Groups were compared with by one-way ANOVA with Tukey's multiple comparison test. **i** Representative images of chromosomal abnormalities quantified in lymphoblast metaphases. **j**–**m** Quantification of chromosomal abnormalities per metaphase in lymphoblast cell lines, control $n = 20$, proband $n = 22$. Each point is a single metaphase, only the mean is shown. Groups were compared with a Mann–Whitney test. **n** Mean telomere intensity from the same samples as (**j-m**), only the mean is shown. Groups were compared with a Mann-Whitney test. For (**a**), (**b**), (**e**), and (**f**), groups were compared by one-way ANOVA with Dunnet's multiple comparison test. For all comparisons, ns, non-significant, $p \geq 0.05$ and ****$p < 0.0001$. Source data are provided as a Source Data file.

processive C50/56A led to a greater proliferation deficit (Supplementary Fig. 5c, d). Prior work has suggested that endogenous TR can rescue cells from the effects of variant telomere sequences[12]. Thus, cells likely have a mixture of variant and canonical sequences, but the precise proportion is unclear.

To more directly examine whether variant repeats cause a DDR, we analyzed telomere dysfunction-induced foci (TIFs) in hTERT-RPE cells over-expressing variant telomerase RNAs. Cells transduced with C50/56A, but not C50A, had a significant increase in TIFs, with 23% of cells having 5 or more TIFs (Fig. 4d, e). Cells transduced with AU5 had

fewer TIFs, but had dysmorphic and enlarged nuclei likely due to widespread chromosome fusions caused by this sequence[34] (Fig. 4f). We additionally quantified chromosome abnormalities, including chromosome fusions, fragile telomeres, signal-free ends, and missing sister telomeres (Fig. 4i), but found no significant chromosomal deviations (Supplementary Fig. 5g–j). Thus, over-expression of a processive version of the variant template was sufficient to cause a DDR, but non-processive addition appears to be tolerated.

We next tested if there were elevated levels of DDR within a system where both alleles of *TERC* were expressed at approximately physiologic levels. We made several attempts to generate genome-edited isogenic cell lines using CRISPR/Cas9, but were unsuccessful. We therefore generated a lymphoblast cell line from a PBMC sample from the proband and compared this to an age-matched control lymphoblast cell line. We examined the number of 53BP1 foci in control and patient-derived lymphoblasts and found no differences in the number of 53BP1 foci or nuclear size (Fig. 4g, h, Supplementary Fig. 5f) suggesting that physiologic expression of the variant TR in the context of the WT TR was not sufficient to drive a DDR. We examined metaphase spreads from the control and patient-derived lymphoblasts to determine if the variant TR may cause chromosomal abnormalities. This analysis showed a small but significant increase in the number of signal-free ends, missing sister telomere foci, chromosomal fusions (Fig. 4i–l), and a small decrease in the number of fragile telomeres (Fig. 4m) in patient-derived lymphoblasts. Consistent with these abnormalities, we observed significantly shorter telomeres in the patient-derived lymphoblasts (Fig. 4n). Given that over-expression of the variant sequences was not sufficient to cause chromosomal dysfunction, short telomeres likely mediated the phenotypes observed in the patient derived lymphoblasts. However, because the above experiments were performed in transformed cells, we cannot exclude that transformation disrupted some aspects of the DNA damage response.

### Variant telomere sequence and telomere length regulation

We hypothesized that the dramatic decrease in RAP caused by the C50A variant would lead to significantly shortened telomeres in individuals with this variant[26]. We measured the proband's telomere length using flow cytometry coupled with fluorescent in situ hybridization (FlowFISH) using peripheral blood mononuclear cells (PBMCs) collected six years after lung transplantation. The proband's lymphocyte and granulocyte telomere length fell near the 10th percentile which is consistent with the expected range for a patient with IPF[35] (Fig. 5a, b). No PBMC sample was available for the proband's son. The variant sequence present in the proband's telomeres should not bind to the telomere probe and may therefore lead to an underestimate of the proband's true telomere length.

To investigate the influence of a variant telomere sequence on telomere length regulation, we examined interphase and metaphase chromosomes using FISH on lymphoblasts derived from the probands PBMCs (Supplementary Fig. 7a–e). Besides a weaker telomere FISH signal from patient-derived lymphoblasts, other characteristics were similar to control lymphoblasts (Fig. 4n, Supplementary Fig. 7a–j). We hypothesized that the addition of variant telomere sequence may influence length regulation at individual telomeres. To address this question, telomere foci were imaged and normalized to the mean telomere intensity per cell. The normalized distributions of telomere lengths were similar in control and patient-derived cells (Fig. 5c). For both cell lines, telomeres in interphase had a skewed distribution towards bright telomeres, while telomeres in metaphase had a roughly normal distribution, likely due to telomere cluster in interphase cells (Fig. 5c). Interestingly, the signal from the canonical telomere sequence probe (TTAGGG) was brighter when co-localized with a variant sequence (Fig. 5d). In both interphase and metaphase cells, the variant-associated telomeres skewed significantly rightward compared

to telomeres that were not associated with a variant sequence, indicating that the variant telomeres were much more likely to be associated with longer telomeres (Fig. 5d–e).

We next analyzed how the variant telomerase influenced the 3′ telomere terminal sequence. We collected genomic DNA from the proband lymphoblast cell line and sequenced the 3′ telomere termini using C-tailing and next-generation sequencing (Fig. 5f). As described above, lncRNA-seq of patient-derived PMBCs demonstrated slightly higher expression of the variant TR (Supplementary Fig. 3c). Despite the imbalance in expression, the vast majority of the proband's telomeres terminated in the wild-type telomere sequence (Fig. 5g). We also analyzed which permutation of the wild-type and variant telomere sequence occurred at the telomere terminus. As previously reported, telomeres terminating in a wild-type telomere sequence most frequently ended in the permutation -GGTTAG, matching the telomerase catalytic cycle[36,37]. Telomeres terminating in the variant telomere sequence most frequently terminated in -GGTTTA, corresponding to the pause opposite rU47 identified in our biochemical characterization (Fig. 5h).

### Generation of highly extendable telomeres

We hypothesized that the variant template TR may have two opposing effects on telomere length. Our analysis above demonstrated that the presence of the variant significantly decreased telomerase RAP and therefore limited the capacity of telomerase to extend telomeres. However, the variant telomere sequence could disrupt telomere length regulation by preventing binding of shelterin. The POT1-TPP1 heterodimer is an important regulator of telomere length and processivity, and previous experiments have shown that binding of POT1 to the 3′ terminus of the telomere can prevent telomerase access[38]. Moreover, removal of the DNA binding domain of POT1 (POT1ΔOB) results in dramatic telomere lengthening[9]. We hypothesized that POT1 would be unable to bind the variant telomere sequence, allowing telomerase greater access for telomere extension, somewhat mitigating the loss in RAP[3] (Fig. 6a).

We tested if POT1 could bind the variant sequence in vitro by EMSA and found that its affinity was greatly reduced, although not entirely abolished (Fig. 6d, Supplementary Fig. 8a–c). We next tested if the variant sequence could prevent POT1-mediated inhibition of telomerase activity in vitro. Oligonucleotides containing the canonical telomere sequences were not extended by telomerase when preincubated with POT1. However, oligonucleotides containing the variant sequence required higher concentrations of POT1 during preincubation to inhibit telomerase activity (Supplementary Fig. 8d–f). The oligonucleotides used in this experiment contain a POT1 binding site 3′ to the variant sequence as this is necessary for POT1 to enhance telomerase RAP[5] (Supplementary Fig. 8e, f).

We showed previously that C50/56A TR extended telomeres similarly to WT TR (Fig. 3d, e) despite the significant defect in RAP (Fig. 2). We hypothesized that when a variant sequence is incorporated into a telomere it decreases POT1 binding, which in turn increases telomerase access, partly compensating for the decreased RAP (Supplementary Fig. 9a, b). To test if increased telomerase access with the variant telomere sequence leads to the equivalent addition by WT and C50/56A TR, we overexpressed POT1ΔOB. In this setting, WT TR added nearly double the amount to the telomere as C50/56A TR (Fig. 6c, Supplementary Fig. 9c). C50/56A TR was moderately enhanced by the addition of POT1ΔOB, but not nearly to the extent of WT. These results support that incorporation of a variant sequence would prevent binding of POT1 and thus create highly extendable telomeres extended by telomerase in a manner analogous to mutations in the DNA binding domain of POT1 and may partly compensate for reduced RAP of the variant telomerase.

## Discussion

We identified a unique heterozygous variant in *TERC* in a patient with IPF that encodes for a variant telomere sequence. The variant was

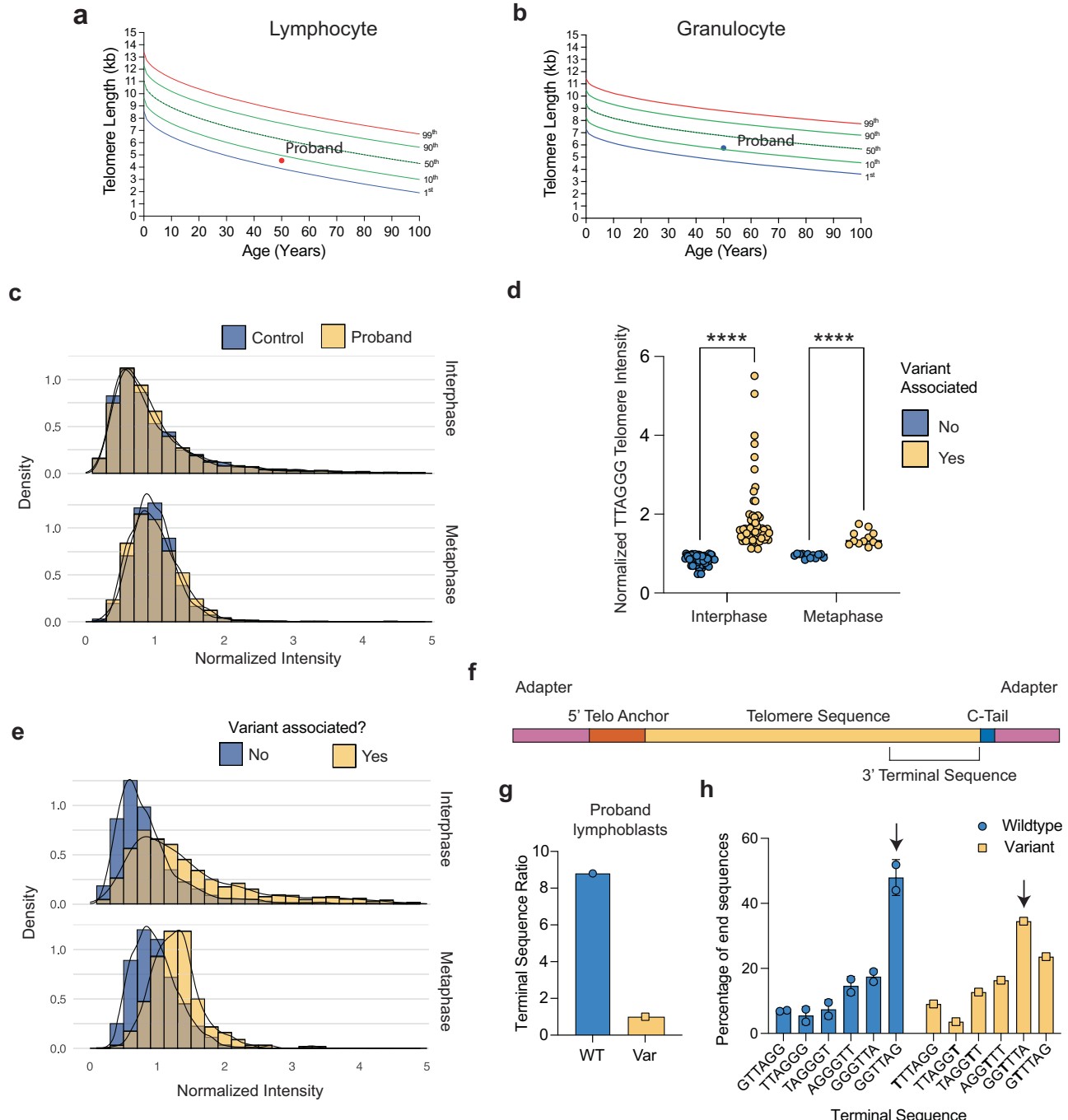

**Fig. 5 | Telomere length and dynamics caused by a variant telomere sequence.**
**a**, **b** FlowFISH telomere length measured in the proband seven years post-lung transplantation. Telomere length is shown relative to a nomogram of leukocyte telomere lengths from healthy controls. Lymphocyte (**a**) and granulocyte (**b**) telomere length are shown. **c** Telomere intensity distribution for individual telomeres for the control and proband cell lines for both interphase (top) and metaphase (bottom) chromosomes. Telomeres are normalized to the mean telomere intensity within each cell. **d** Average intensity of wild-type telomere probe for wild-type telomeres either associated or not associated with a variant telomere, averaged per cell ($n = 61$) or metaphase ($n = 12$). Intensity was normalized the same as (**c**). Groups were compared with Mann–Whitney test and the mean is shown. **e** Telomere intensity distribution for individual telomeres either associated or not associated with variant telomere foci for both interphase (top) and metaphase (bottom) chromosomes. Normalization was the same as (**c**). **f** Schematic showing the C-tailed PCR product for sequencing of the terminal telomere. **g** The percentage of telomeres that terminate in a wild-type vs variant sequence. **h** Occurrence of each permutation as a fraction of total telomere ends of the wild-type or variant sequence. The arrow indicates the permutation expected based on the biochemically predicated pause site for both wild-type and variant TR. Data for the wild-type sequences were determined from both the control and patient-derived lymphoblasts whereas the data for the variant sequence could be determined only for the patient-derived lymphoblasts. ****$p < 0.0001$. Source data are provided as a Source Data file.

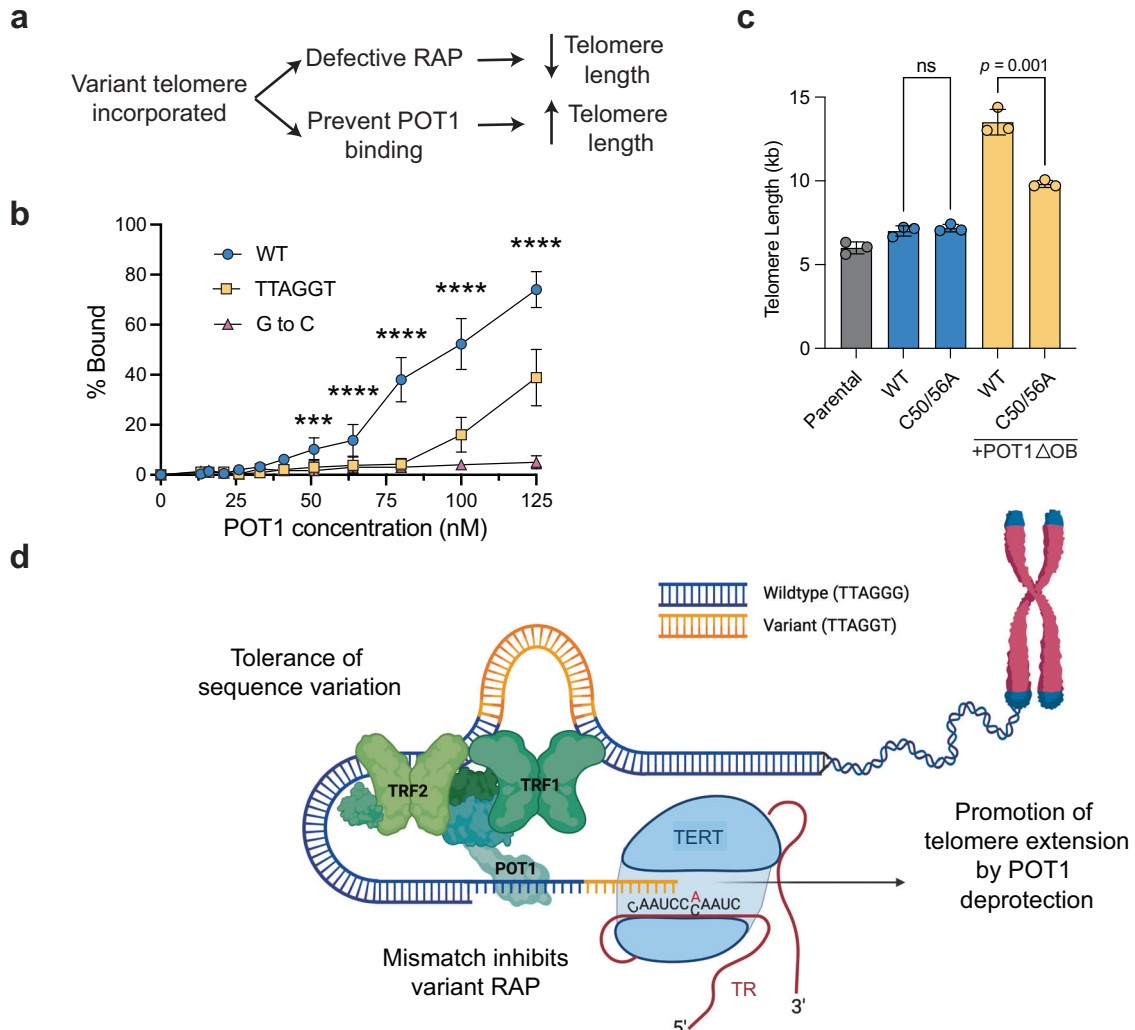

**Fig. 6 | Proposed mechanism for tolerance and telomere length maintenance.**
**a** Graphic describing the opposing effects a variant telomere sequence has on telomere length maintenance. **b** A binding curve for POT1 generated from electromobility shift assays (EMSAs) for POT1 with single-stranded DNA oligonucleotides corresponding to either the G-strand wild-type telomere sequence (WT; $n = 6$), the variant TTAGGT sequence (C50A; $n = 9$), or a G to C substitution of the wild-type sequence[5] (TTAGCG; $n = 3$). The latter is known to inhibit POT1 binding and serves as a negative control. Mean ± s.d. is shown. Statistical comparison between WT and TTAGGT binding is shown, groups are compared with two-way ANOVA with Tukey's multiple comparison test. **c** Quantification of TRF Southern blot of HCT116 cells 13 days post-transduction. Cells were co-transduced with WT or C50/56A TR and POT1ΔOB. $n = 3$ for each group. Mean ± s.d. is shown, WT and C50/56A were compared by a two-tailed unpaired $t$ test. **d** Model showing how shelterin might flexibly bind around the variant telomere sequence to effectively protect the telomere, while variant telomere sequence at the telomere end primes it for telomerase elongation due to decreased POT1 binding, attenuating the effect of reduced RAP from the variant sequence. The graphic was created with BioRender.com and released under a Creative Commons Attribution-NonCommercial-NoDerivs 4.0 international license. ns, non-significant, ***$p < 0.001$, ****$p < 0.0001$. Source data are provided as a Source Data file.

tolerated sufficiently to permit the proband to live disease-free for at least 4 decades and be inherited across at least one generation. One prior published instance of a germline template variant (r.48a>g) was reported in a patient with dyskeratosis congenita, a severe bone-marrow failure syndrome caused by extremely short telomeres, and molecular characterization determined that the variant TR resulted in a non-functional telomerase[39–41]. Both family members identified here had significant variant telomere sequences within their genomes supporting that telomerase loaded with the variant TR template is active in vivo. We found that the proportion of variant telomere sequence increased significantly across one generation, from 2.4% to 9.1%, suggesting that variant telomeres were added in the proband's germline or early during the development of the proband's son. Despite having a significant proportion of variant telomere sequence, the proband's son reported no significant medical conditions, demonstrating that telomeres can tolerate significant degeneracy and remain functional, at least through adolescence.

While telomere sequence degeneracy has been recognized for decades[42], this is the first report of a germline-encoded variant that is inherited across at least one generation. Variant telomere repeats have largely been investigated in the context of alternative lengthening of telomeres (ALT), a telomerase-independent method of telomere maintenance found in some cancers that enriches for variant telomere repeats via amplification and recombination of sub-telomeric sequences[28,30]. In addition, sequence-altering telomerase template variants, resulting in variant telomere addition, have been identified in tumors[43]. Variant telomere repeats are enriched in tumor-derived cells potentially due to rapid proliferation, chromosomal rearrangements that can amplify sub-telomeric regions, and defects in the DDR[44]. Recently, the development of long-read sequencing has revealed significant telomere degeneracy in non-cancerous cells, particularly near sub-telomeric regions of chromosomes[37,45]. Taken together, it appears that telomeres can tolerate significant degeneracy, both distally and proximally, and remain functional.

The shelterin components bind with high sequence specificity to telomeric DNA and are essential for genome stability[2,3]. It is therefore unsurprising that expression of variant telomere sequences are toxic to cells[13–15]. We showed that POT1 bound poorly to the variant sequence identified in this family and that over-expression of the variant was not sufficient to drive a DDR. When we examined lymphoblasts derived from the proband, we also could not identify a DDR response (indicated by 53PB1 staining). However, we did identify several chromosomal abnormalities in the proband-derived lymphoblasts that are consistent with short telomeres and we cannot excluded that a DDR related to a variant did not contribute to the genome instability. Over-expression of a modified TR that was more processive (C50/56A) did limit cellular proliferation and cause a DDR. The differing responses are likely driven by the number of variant sequences that is incorporated into the telomere. Ultimately, examining telomere dynamics in cells that endogenously express telomerase (i.e. pluripotent stem cells) would be informative. Under endogenous conditions, both the variant- and WT-TR are co-expressed, and telomeres likely contain a mixture of canonical and non-canonical repeats that are sufficient for shelterin to bind and suppress the DDR. Tolerance of variant sequences is likely linked to flexible shelterin binding—both TRF1 and POT1 can bind to telomeric sequence interspersed with non-telomeric sequence in vitro[2,46]. The spatial flexibility of these subunits of the shelterin complex provides a mechanism by which telomeres could withstand some degenerate sequence so long as sufficient wild-type telomere is present (Fig. 6d). The studies we performed are limited by the use of cell lines that may have lost several checkpoints during the immortalization procedure. Nonetheless, the variant has persisted for two generations in a family and one individual reported no significant medical concerns despite significant degeneracy in his telomeres. Together, these data suggest that so long as some minimum wild-type sequence is present, the telomeres can contain the TTAGGT sequence without significant functional consequences.

Several variant telomere sequences have been used to measure nascent telomere extension in vivo[16,47,48]. The sequence identified here is an alternative and may be preferred as it arose spontaneously, is compatible with long-term survival of some cell types, and retains some affinity for POT1. We demonstrated that variant telomere sequences can be a powerful tool for measuring in vivo telomerase activity and enzyme kinetics using FISH and sequence analysis. The use of variant sequences to study telomerase activity is not without caveats. Our findings demonstrate that altered telomere repeats can dramatically alter telomerase activity by altering shelterin binding and influencing cellular fitness. Thus in vivo measurement of processivity may be influenced by factors external to enzyme kinetics (i.e., changes in binding of shelterin or selection against cells that incorporate long tracks of variant repeats). As telomere sequencing tools further develop, including long-read sequencing, the incorporation of variant sequences may emerge as a tool for understanding telomerase activity and telomere length regulation in intact cells.

Our analysis of telomerase activity and telomere length suggests a complex mechanism of telomere maintenance for this variant. Telomerase containing C50A TR results in almost complete loss of RAP (Fig. 2). This loss in RAP was primarily mediated through a mismatch between the nascent variant telomere and r.56C of TR. Correcting this mismatch by the creation of the C50/56A variant substantially increased RAP and allowed for additional stimulation by POT1-TPP1, although C50/56A was still much less processive than WT. Paradoxically, when we over-expressed C50/56A TR, it extended telomere equally as well as WT TR, despite the deficit in RAP. Moreover, we examined telomeres using FISH, we noted that when WT telomere foci co-localized with variant foci, they were often more intense than foci that did not. Based on these observations, we hypothesized that variant telomere sequences may function mechanistically similar to POT1ΔOB and prevent shelterin-mediated inhibition of telomerase

addition. Indeed, we found that in vitro telomerase direct assays were not inhibited by pre-incubation with POT1 when oligos composed of the variant sequence were used. Moreover, when POT1ΔOB was co-expressed in cells, C50/56A TR was unable to enhance telomere lengthening to the same extent as wild-type TR. Finally, when we examined the 3′ terminal telomere sequence in lymphoblasts derived from the proband, the overwhelming majority of telomeres terminated with the canonical sequence despite expression of both TR alleles. Previous studies in the budding yeast *Kluyveromyces lactis* found that specific mutations that alter the telomeric DNA sequence caused dramatic telomere lengthening[49] and mammalian cells that co-overexpress wild-type and variant TR have longer telomeres than those that over express wild-type TR alone[29]. Taken together, these data suggest that when variant telomere sequences are incorporated, they can have conflicting effects on telomerase activity and telomere length regulation.

Telomere-mediated disease in humans results from short telomeres limiting the capacity of stem/progenitors to proliferate. The proband in our study had a clinical presentation consistent with individuals with short-telomere-mediated disease. However, his clinical course was atypical in that his telomeres were only moderately shorter than expected for his age and he has tolerated 7.5 years of full immune suppression post lung transplantation[50–55]. Cellular dysfunction in the individuals we identified could potentially be mediated by two independent mechanisms—short telomeres or telomere uncapping caused by a variant telomere sequence[12,14,15,24]. We did not see any gross DNA damage effects in our cell-based assays using the patient variant, even when overexpressed, supporting tolerance of the C50A TR variant. Therefore, disease in this individual is likely mediated by short telomeres that are attenuated by the mechanisms discussed above. In addition, our findings point to a novel function of shelterin in promoting elongation of telomeres that have incorporated variant sequences (Fig. 6d). This study informs a more nuanced understanding of how both telomere composition and length can influence health and disease.

## Methods

### Study approval

All studies included here were approved by the University of Pittsburgh Institutional Review Board (STUDY 20060250 and STUDY18070008). All research subjects gave written informed consent and minors assented to study participation. No subjects were compensated for participation in this study. All studies were performed in accordance with the criteria set by the Declaration of Helsinki.

### Telomere sequence composition

Whole genome sequencing was performed as described previously[27]. Raw fastq files were used for all analysis. Control samples used in this study were lung transplant recipients that did not have any rare genetic variants in telomere-related genes. Sequencing reads were deemed telomeric reads if they had a total of 13 telomere repeats (TTAGGG, TTAGGT, CCCTAA, or ACCTAA) anywhere within the 150 base pair read. We noted that read quality of telomeric reads dropped significantly after the first 60 base pairs and all reads were trimmed accordingly (Supplementary Fig. 1b, c). Total wild-type and variant telomere sequence was quantified and normalized to the total sequence length using Perl scripts (see source data). Additionally, the numbers of consecutive wild-type and variant repeats were also quantified and normalized to the total percentage of wild-type or variant repeats using the same approach.

### Terminal telomere sequencing

Genomic DNA was collected from the lymphoblasts derived from the proband. DNA was C-tailed by combining 2 μg of DNA, 10U Terminal

Transferase (NEB # M0315S), 1x Terminal Transferase Buffer (NEB # M0315S), 1x $CoCl_2$ ((NEB # M0315S), and 5 mM dCTP at 37 °C for 1 h followed by 70 °C for 10 min. The reaction was then repaired using Sulfolobus DNA Polymerase IV (NEB #M0327S), 1x Thermopol Buffer (NEB #B9004S), 200 µM dNTPs, and the Tel End Seq Repair – Nextera MM and HW primers in a 1:2 ratio at a final concentration of 0.5 µM. The reaction was incubated for 5 min at 60 °C and then 15 min at 65 °C. 1 µl of each RsaI (NEB #R0167S) and HinfI (NEB # R0155S) were added to the reaction and it was incubated at 37 °C overnight. The DNA was purified on a silica column. Terminal telomere DNA was then amplified by PCR. PCR was conducted with Herculase II Polymerase (Agilent #600675), 1x Herculase Buffer (Agilent #600675), 1 M betaine, 200 µM dNTPs, and 0.5 µM of the primers PCR1 Fwd Nextera 1-1 or 1-2 and PCR1 Rev Nextera 2. The PCR was conducted for 20 rounds with a 98 °C−30 s denaturation, 52 °C−30 s annealing, and 72 °C−3-min annealing step. The PCR product was run on an agarose gel and the smeared DNA at 300-500 base pairs was gel extracted. PCR 2 was then performed under similar conditions to PCR1 to add Illumina sequencing and index adapters (PCR2 Fwd S502 and S503; PCR2 Rev N701 and N702). PCR2 was run for 15 rounds with 98 °C−30 s denaturation, 57 °C−30 s annealing, and 72 °C−1-min annealing step. The PCR product was gel extracted for bands in the 300-500 base pair range. Libraries were quantified with an Agilent TapeStation 4150 and Qubit fluorometer prior to sequencing on a MiSeq Micro v2 300 flow cell (MS-103-1002), 2x150bp. The library pool was loaded at 10 pM with 30% PhiX for read diversity and 30% Illumina DNA Prep RNA-seq library pool for read diversity and index diversity. DNA sequences from the reverse read (starting at the C-tail) were filtered for quality by selecting reads with ≥5 CTAACC or CTAAAC repeats, containing 6 or more sequential G nucleotides, and no undefined bases for 12 nucleotides after the string of Gs. The percentage of wild-type and variant telomere sequence adjacent to the C-tail and their permutation were quantified using a custom python script.

## Cell culture procedures

LOX Melanoma cells were a gift from Dr. Bradley Stohr and cultured in RPMI (ThermoFisher, Gibco) supplemented with 10% fetal bovine serum (ThermoFisher, Gibco), penicillin (120 u/mL), streptomycin (100 µg/mL), and L-glutamine (2 mM). Cells were maintained at 37 °C and 5% $CO_2$ in a humidified incubator. HEK293FT (ThermoFisher; R70007) and HCT116 (ATCC® CCL-247™) cells were cultured in DMEM (ThermoFisher, Gibco) supplemented with 10% fetal bovine serum (ThermoFisher, Gibco), penicillin (120 u/mL), streptomycin (100 µg/mL), and L-glutamine (2 mM) and maintained at 37 °C and 5% $CO_2$ in a humidified incubator. hTERT-RPE-1 (hTERT-RPE) cells were a gift from Dr. Patricia Opresko and cultured in DMEM-F12 (ThermoFisher, Gibco) supplemented with 10% fetal bovine serum (ThermoFisher, Gibco), penicillin (120 u/mL), streptomycin (100 µg/mL), and L-glutamine (2 mM) and maintained at 37 °C and 5% $CO_2$ in a humidified incubator. The identities of LOX Melanoma, HCT116, and hTERT-RPE cells were verified using STR profiling (ATCC). The control lymphoblast cell line was COLO 829BL from ATCC. Both control and patient lymphoblast cell lines were maintained in RPMI (ThermoFisher, Gibco) supplemented with 10% fetal bovine serum (ThermoFisher, Gibco), penicillin (120 u/mL), streptomycin (100 µg/mL), and L-glutamine (2 mM) and maintained at 37 °C and 5% $CO_2$ in a humidified incubator.

## Lymphoblast cell line generation

A lymphoblast cell line was generated from the proband's PBMCs by Epstein-Barr Virus (EBV) transformation. The proband PBMCs were incubated with Human gammaherpesvirus 4 (HHV-4; also known as Epstein-Barr Virus) obtained from ATCC (VR-1492) and irradiated MRC-5 feeder cells in RPMI (ThermoFisher, Gibco) supplemented with 10% fetal bovine serum (ThermoFisher, Gibco) according to the ATCC

protocol (https://www.atcc.org/resources/technical-documents/lymphocyte-transformation-using-atcc-vr-1492).

## Creation of TR variants

Lentiviruses expressing template-modified TR were created using the pLV-IU1-hTR-CMV-GFP vector (a generous gift from Dr. Bradely Stohr). Changes to the 11 base pair template and to the selection marker were made via restriction digest followed by Gibson Assembly of synthesized fragments (Integrated DNA technologies). The new templates for each TR are as follows, WT: 3'-CAAUCCCAAUC-5'; C50A: 3'- CAAUC-CAAAUC-5'; C50/56 A: 3'-AAAUCCAAAUC-5'; AU5: 3'-AAUAUAUAUAU-5'; TSQ1: 3'-CCAACGCCAAC-5'. Template sequences for AU5 and TSQ1 were derived from previous studies[16,34]. Vectors were generated with expression cassettes for GFP, Puromycin selection, and Blasticidin selection. The empty vector (EV) was created by excising the entirety of the TR sequence from the plasmid. All changes were verified by Sanger Sequencing through GENEWIZ (subsidiary of Azenta Life Sciences, South Plainfield, New Jersey) or Plasmidsaurus (Eugene, Oregon).

## Lentiviral production

Lentivirus was packaged in HEK293FT cells using the pLV-IU1-hTR-CMV-GFP, -Puro, or -Blast vector plasmid, the Delta 8.9 packaging plasmid, and the VSV.G envelope plasmid as previously described[56]. The media of the 293-FT cells was replaced with DMEM with 1% BSA without antibiotics and the cells were transfected with the appropriate plasmids and polyethylenimine (PEI) at a ratio of 2 µg:1µg in Optimem. Supernatant was collected at 48 and 72 h, filtered using a 0.45 µm sterile filter (Thermo Fisher), and concentrated using Amicon Ultra Centrifugal Filters (15 mL, Millipore Sigma). GFP lentivirus was titered in 293-FT cells using flow cytometry. Following concentration, lentivirus was stored at −80 °C.

## PNA probes

All PNA probes were synthesized at PNABio (Thousand Oaks, California). The WT telomere probe was Cy3 conjugated (TelC-Cy3, F1002) repeats of 5'-CCCTAA-3'. The pan-centromere probe was Alexa 488 conjugated (CENPB-488, F3004) 5'-ATTCGTTGGAAACGGGA-3'. The variant telomere probe was custom made and conjugated to Alexa 647 5'-ACCTAAACCTAAACCTAA-3'.

## Proliferation assay

LOX Melanoma and hTERT-RPE cells were transduced with the WT, C50A, C50/56A, AU5, and TSQ1 hTR lentivirus co-expressing GFP at an MOI of 5 in the presence of 8µg/mL polybrene. Media was replaced with the standard media either 5 h (hTERT-RPE) or 24 h (LOX Melanoma) post-transduction. Two days post-transduction, 400 cells were plated per well in a 6-well plate. After 7 (LOX Melanoma) or 9 (hTERT-RPE) days the cells were imaged on a ChemiDoc MP Imaging System (Bio-Rad) for GFP-positive colonies. The cells were then washed with PBS and fixed with ice-cold methanol, followed by staining with Crystal Violet stain in 25% methanol. The plates were imaged again on a ChemiDoc MP Imaging System (Bio-Rad) and the percent area of the plate covered for both GFP and Crystal Violet was quantified using FIJI (Image J version 2.1.0/1.53c).

## Competition assay

LOX Melanoma and hTERT-RPE cells were transduced with 8µg/mL polybrene at an MOI of 0.5 with the WT, C50A, C50/56A, AU5, or TSQ1 hTR lentivirus co-expressing GFP. The percentage of GFP-positive cells was measured 48- or 72-h post-transduction, and every 2-4 days after via flow cytometry (BD LSR Fortessa) and analyzed by FlowJo (Version 10.7.1). GFP percentage at later days was normalized to the first time-point. Gating strategy is displayed in Supplementary Fig. 6.

## Lung telomere fluorescent in situ hybridization (FISH)

Lung sections from the proband's explanted lungs and donor lungs that were deemed not suitable for transplantation (control patient) were analyzed. Formalin-fixed, paraffin-embedded sections were warmed to 65 °C and then hydrated by sequential incubation in xylenes and ethanol following standard protocols. The slides were placed in an EasyDip™ Slide Staining Jar and incubated in antigen unmasking solution (H-3300; Vector Laboratories) for 30 min in a steamer. The slides were then cooled, washed in DI water and underwent ethanol dehydration (70-90-100 percent EtOH) and then fully dried. 35 µl of PNA hybridization buffer (70% formamide [v/v], 0.25% [w/v] blocking reagent, 10 mM Tris-Cl, pH 7.5) and the previously described PNA probes for the wild-type (0.3 µg/mL) and variant (0.6 µg/mL) were added to the slides and denatured at 85 °C for 5 min. Slides were hybridized at room temperature for 2.5 h in a humid chamber. After hybridization, they were washed two times for 15 min in PNA Wash Buffer A (70% Formamide, 10 mM Tris-Cl pH 7.5) and three times for 5 min in PBS with 0.5% Tween-20. Slides were incubated with DAPI (500 ng/mL) for 10 min, and then washed with DI water, drained, and mounted with ProLong™ Diamond Antifade Mountant (Invitrogen). Slides were stored at 4 °C until they were imaged at 60X on a Nikon ECLIPSE N*i*-E fluorescent microscope. The wild-type and variant telomeres were quantified using Nikon Elements GA3. Variant telomeres were classified as foci that co-localized with wild-type telomeres within the nucleus.

## Metaphase spread and FISH protocol

This protocol is similar for all cell types, with slight variation in time in the hypotonic solution. Metaphases were conducted as described as previously published[57]. In brief, when cells were at ~50% confluency they were treated with 0.1 µg/mL colcemid (KaryoMAX colcemid, Gibco) for 2 h to arrest cells in metaphase. Adherent cells were then trypsinized and collected in a 15 mL falcon tube whereas suspension cells were directly collected in a 15 mL falcon tube. Cells were treated with a 75 mM KCl hypotonic solution for 8–10 min at 37 °C. They were then placed on ice and an ice-cold 3:1 methanol to glacial acetic acid solution fixative was added to a total of 10% v/v of fixative to hypotonic solution. The cells were centrifuged (185*g*), and then resuspended in cold fixative, prior to centrifuging again. The suspended cells were dropped from a height of approximately 3 feet onto slides that had been soaked in cold methanol and dipped in ice water. The slides were streamed for approximately 3 s before sitting on a wet paper towel for 5 min, and then allowed to dry overnight. Slides were checked under a light microscope to ensure metaphase spreading before continuing. Slides were placed in EasyDip™ Slide Staining Jars and rehydrated for 5 min in PBS at 37 °C, fixed with 2% PFA at room temperature for 5 min, washed 3x in PBS, and then treated with 100µg/ml RNaseA (Qiagen) in PBS at 37 °C for 15 min. Following this, slides were treated with 0.2 mg/mL pepsin (MP Biomedicals) in a 0.1 M HCl (pH ~2) solution at 37 °C for 15 min, fixed again with 2% PFA at room temperature for 5 min, and then washed 3x with PBS. Slides were dehydrated using a 70-90-100 ethanol grade and allowed to dry to completion. Wild type, variant, and CenpB probes were prepared into a single tube at a concentration of 0.5µg/mL per probe in PNA hybridization solution (70% formamide [v/v], 0.25% (w/v) blocking reagent, 10 mM Tris-Cl, pH 7.5). 40µl of the probe solution was added per slide with a glass coverslip, heated on a heat block to 83 °C for 5 min, and then allowed to hybridize in a humid chamber overnight at 4 °C. The following morning slides were washed twice for 15 min with PNA Wash A solution (70% [v/v] deionized formamide and 10 mM Tris-Cl, pH 7.5) and then three times for 5 min with PNA wash B solution (50 mM Tris-Cl, pH 7.5, 150 mM NaCl, 0.1% [v/v] Tween 20). DAPI added at a concentration of 500 ng/mL to the second wash. Slides were again dehydrated using a 70-90-100 ethanol grade and allowed to dry to completion before mounting with ProLong™

Diamond Antifade Mountant. Slides were imaged and analyzed as described above.

## Immunofluorescence with telomere FISH

hTERT-RPE cells were transduced with pLV-IU1-hTR-CMV-GFP at an MOI of 5. At 6 days for hTERT-RPE 150,000 cells were plated onto sterilized glass coverslips on a 12-well plate. 24 h after seeding the cells were washed and fixed with 4% paraformaldehyde in PBS for 5 min. Lymphoblast cells were collected at a concentration of 400,000 cells/mL. 200 µL were spun onto glass slides using a cytocentrifuge (Shandon CytoSpin 4). The slides were dried overnight and fixed with 4% paraformaldehyde in PBS for 5 min. For all cell types, cells were then permeabilized for 5 min at room temperature with KCM (120 mM KCl, 20 mM NaCl, 10 mM Tris, 0.1% triton, pH 7.5) followed by washing twice for 10 min in PBS, and then twice for 10 min in IF Wash (PBS with 0.5% Tween-20 and 0.25% BSA). Cells were then blocked for 30 min with IF Blocking (PBS with 0.5% Tween-20, 2.5% BSA, and 10% Goat Serum). A 1:400 dilution of 53BP1 antibody (NOVUS Biologicals) in IF Blocking solution was added to the coverslips and allowed to hybridize overnight at 4 °C. The coverslips were washed with IF Wash and the secondary antibody, Goat anti-Rabbit IgG (Invitrogen) added at 1:1000 added at 1:1000 in PBS-T (PBS, 0.1% Tween-20) for 1 h. Cells were washed in PBS-T and fixed with 4% PFA. They were then wash with DI water and ethanol dehydrated (70-90-100 percent graded ethanol series) and fully dried. FISH for the wild-type telomere sequence was performed as described above. Telomere-induced foci (TIFs) were defined as 53BP1 foci that colocalized with telomeres and quantified with Nikon Elements GA3.

## FISH on cultured cells

hTERT-RPE cells were transduced with pLV-IU1-hTR-CMV-GFP at an MOI of 5. Lox Melanoma cells were transduced with pLV-IU1-hTR-CMV-Puro for long-term culture. At 6 days (for hTERT-RPE) and 4 and 6 days (LOX Melanoma) cells were plated, fixed, and permeabilized as described in the immunofluorescent section. After permeabilization, cells were dehydrated and hybridized to both the wild-type and TTAGGT PNA probes using the method described. Cells were imaged at 60X on a Nikon ECLIPSE N*i*-E fluorescent microscope. Wild-type and variant telomeres were quantified using Nikon Elements GA3 and variant telomeres were classified as foci that co-localized with wild-type telomeres within the nucleus.

## FlowFISH telomere length measurement

FlowFISH was conducted on the proband's PBMCs six years post-transplant at the Molecular Diagnostics Lab at Johns Hopkins.

## Telomerase preparation

HEK-293T (ATCC) cells were grown to 90% confluency in Dulbecco's modified Eagle's medium (Gibco) supplemented with 10% FBS (Hyclone) and 1% penicillin-streptomycin (Corning) at 37 °C and 5% $CO_2$. Cells were transfected with 10 µg of pSUPER-hTR plasmid (or with C50A or C50/56 A mutations) and 2.5 µg of pVan107 hTERT plasmid diluted in 625 µl of Opti-MEM (Gibco) mixed with 25 µl of Lipofectamine 2000 (ThermoFisher) diluted in 625 µl of Opti-MEM. Cells expressing hTR and 3xFLAG-tagged human hTERT were harvested 48 hr post-transfection, trypsinized and washed with PBS, and then lysed in CHAPS buffer (10 mM Tris-HCl, 1 mM $MgCl_2$, 1 mM EDTA, 0.5% CHAPS, 10% glycerol, 5 mM β-mercaptoethanol, 120 uM RNasin Plus (Promega), 1 µg/ml each of pepstatin, aprotinin, leupeptin and chymostatin, and 1 mM AEBSF) for 30 min at 4 °C. 80 µL of anti-FLAG M2 bead slurry (Sigma) (per T75 flask) was washed three times with 10 volumes of 1X human telomerase buffer (50 mM Tris-HCl, pH 8, 50 mM KCl, 1 mM MgCl2, 1 mM spermidine and 5 mM β-mercaptoethanol) in 30% glycerol and harvested by centrifugation for 1 min at 1500*g* and

4 °C. The bead slurry was added to the cell lysate and nutated for 4–6 h at 4 °C. The beads were harvested by 1 min centrifugation at 1500*g*, and washed 3X with 1X human telomerase buffer with 30% glycerol. Telomerase was eluted from the beads with a 2x the bead volume of 250 µg/mL 3X FLAG® peptide (Sigma Aldrich) in 1X telomerase buffer containing 150 mM KCl. The bead slurry was nutated for 30 min at 4 °C. The eluted telomerase was collected using Mini Bio-Spin® Chromatography columns (Bio-Rad). Samples were flash-frozen and stored a −80 °C.

### End-labeling of DNA primers
50 pmol of PAGE purified DNA oligonucleotides (IDT) (Supplemental Table 1) were labeled with $\gamma^{32}$P ATP (Perkin Elmer) using T4 polynucleotide kinase (NEB) in 1X PNK Buffer (70 mM Tris-HCl, pH 7.6, 10 mM MgCL$_2$, 5 mM DTT) in a 20 µl reaction volume. The reaction was incubated for 1 h at 37 °C followed by heat inactivation at 65 °C for 20 min. G-25 spin columns (GE Healthcare) were used to purify the end-labeled primer.

### Telomerase activity assay
Reactions (20 µl) contained 1X human telomerase buffer, 5 nM of $^{32}$P-end-labeled primer as indicated in the figure legends and dNTPs at cellular dNTP concentrations (c24 µM dATP, 29 µM dCTP, 37 µM dTTP, 5.2 µM dGTP). For experiments with POT1, primers were pre-incubated with 0-500 µM of POT1 for 15 min are room temperature. The reactions were started by the addition of 3 µl of immunopurified telomerase eluent, incubated at 37 °C for 1 h, then terminated with 2 µl of 0.5 mM EDTA and heat-inactivated at 65 °C for 20 min. An equal volume of loading buffer (94% formamide, 0.1XTBE, 0.1% bromophenol blue, 0.1% xylene cyanol) was added to the reaction eluent. The samples were heat denatured for 10 min at 100 °C and loaded onto a 14% denaturing acrylamide gel (7 M urea, 1X TBE) and electrophoresed for 90 min at constant 38 W. Samples were imaged using a Typhoon phosphorimager (GE Healthcare). Percent primer extension was calculated with ImageQuant TL 8.2 by measuring the intensity of each product band and dividing by the total radioactivity in the lane or total products, as indicated in the figure legends. Processivity was calculated as previously described[58,59].

### POT1 purification
Full-length human POT1 was expressed as a SUMOstar-(His)6- POT1 fusion protein in baculovirus-infected SF9 cells (Thermo Fisher Scientific), as previously described[60]. POT1 was first purified using a HisTrap column (GE Healthcare) and eluted with 200 mM imidazole. After tag cleavage with SUMOstar protease (Ulp1 variant, LifeSensors), the protein was further purified by Superdex 200 size-exclusion chromatography (GE Healthcare Life Sciences) in 25 mM Tris-Cl (pH 8) and 150 mM NaCl.

### Electrophoretic mobility shift assay
Variable concentrations (0–250 nM) of POT1 protein were mixed with 0.25 nM Cy5.5-labeled DNA (Supplemental Table 1) in binding buffer (20 mM HEPES-KOH pH 7.5, 100 mM KCl, 0.5 mM MgCl2, 0.5 mM TCEP-HCl, 0.05% v/v IGEPAL Co-630, 8% glycerol, 50 µg/mL bovine serum albumin) and incubated for 30 min at room temperature (22 °C).10 µL was separated on a 4-20% polyacrylamide-TBE gel in 1x TBE buffer. Gels were imaged using the near-IR setting of a Typhoon scanner (GE).

### Processivity from sequencing
We estimated the in vivo telomerase repeat addition processivity from the sequencing data. This method relies on the assumption that mature wild type and variant hTR are available in equal quantities within the cell and have an equal chance of initially extending the telomere. According to this model, there are three possible outcomes following addition of a variant repeat: (1) processive addition where

telomerase with the variant hTR adds a second 6-nucleotide variant repeat, (2) dissociation of telomerase and reassociation with telomerase loaded with the variant hTR, resulting in a second variant repeat, or 3) dissociation of telomerase and reassociation with telomerase loaded with the wild-type hTR, resulting in a wild-type repeat, terminating that chain of consecutive variant repeats (Supplementary Fig. 3d). The probability of processive addition is $P(x)$ and the probability of telomerase dissociation is $1 − P(x)$. Assuming an equal probability of initial addition by either the wild type or r.50C>A hTR, the probability of telomerase disassociating and then reassociating by chance with r.50C>A hTR is $(1 − P(x))*0.5$. The total probability of a subsequent variant addition $P(Add)$, is therefore the summation of probability of processive addition $P(x)$ and reassociation with a variant telomerase $(1 − P(x))*0.5$. Since these are mutually exclusive events, the probability of variant addition is $P(Add) = P(x) + (1 − P(x))*0.5$. For a completely non-processive telomerase, the probability of addition is $P(Add) = 0 + (1 − 0)*0.5 = 0.5$. Therefore, the probability of two consecutive repeats is 0.5, three is 0.25, etc. For a highly processive telomerase, for example with a $P(x) = 0.9$, the probability of consecutive addition would be $P(Add) = 0.9 + (1 − 0.9)*0.5 = 0.95$. By graphing the $\log_2$ of the proportion of repeats that were found in $x$ or greater consecutive repeats and deriving a line of best fit (Fig. 2g, h), we calculated the probability of consecutive addition, and then work backwards to determine the probability of processive addition, P(x), using the formula $2^{slope} = P(x) + (1 − P(x))*0.5$. Using this method, we can derive the value of $P(x)$, the in vivo telomerase processivity.

### Telomere length assay
To measure telomere extension by the variant sequence, HCT116 cells were co-transduced with hTERT-Neo which expresses a codon-optimized hTERT from an EF-1 alpha promoter and Neo expressed from a hPGK promoter and pLV-IU1-hTR-CMV-Puro and -Blast to form the following variants: EV, WT, C50A, and C50/56A. Cells were selected two days after transduction with Neomycin, Puromycin, and Blasticidin for seven days. At day 15 post-transduction, cells were collected. To measure the effect of POT1ΔOB, HCT116 cells were co-transduced with pWZL N-Myc-hPOT1_deltaOB (a gift from Dr. Agnel Sfeir, Addgene plasmid # 166408) and either WT or C50/56A pLV-IU1-hTR-CMV-GFP. Cells were selected two days after transduction with Blasticidin and collected at day 13 post-transduction. For both experiments, genomic DNA extracted by the salting out method. In brief, cells were lysed in a 20% SDS solution, and an ammonium acetate solution added to precipitate the protein. The protein was then centrifuged and pelleted, and supernatant was mixed with isopropanol to precipitate genomic DNA. The DNA was then resuspended with ethanol, pelleted, dried, and resuspended in TE buffer (10 mM EDTA, 10 mM Tris-HCl). DNA concentration was measured on the Qubit 4 fluorometer. Telomere length was measured using the TeloTAGGG™ Telomere Length Assay kit (cat # 12209136001 Roche) according to the manufacturer's protocol. The membranes were imaged using a ChemiDoc MP Imaging System (Bio-Rad). Telomere length was calculated using the online tool WALTER[61].

### lncRNA-seq and analysis
Long non-coding RNA sequencing (lncRNA-seq) was performed at Novogene (Sacramento, California). RNA was isolated from PBMCs using a Qiagen miRNAeasy kit according to the manufacturer's recommendations. Ribosomal RNA depletion, directional libraries, and 150 base pair paired-end sequencing were performed at Novogene. Raw FASTQ files were trimmed for quality and aligned using STAR[62] and the depth of coverage of TR was examined manually to determine the proportion of reads that mapped to each allele.

### Statistical analysis
Proliferation, competition, DNA damage response, telomere length, and telomere FISH addition assays used at least three independently

transduced wells. Telomerase direct activity assay and the POT1 binding assay were at least three independent experiments. Any other information on the number of biologic replicates that were performed for each experiment is included in the corresponding figure legend. Imaging data was analyzed by Nikon GA3 using the same settings for all images or analyzed by hand using a blinded analysis. Graphs show the standard deviation of the three experiments, all final data analysis was done in GraphPad Prism (Version 8.4.1). Analysis of two groups used a two-tailed unpaired $t$ test, with significance determined as $p < 0.05$. All analyses of one variable with more than two groups were ordinary one-way ANOVAs with Dunnet's or Tukey's multiple comparison correction, with significance determined as $p < 0.05$. All analyses of two variables and multiple groups using two-way ANOVAs with Dunnet's multiple comparison correction, with significance determined as $p < 0.0332$ for multiple comparisons.

### Reporting summary

Further information on research design is available in the Nature Portfolio Reporting Summary linked to this article.

### Data availability

A subset of the sequencing data generated in this study has been included in the Source data. The dataset contains only the telomeric reads to protect the privacy of the individuals included in this study. The raw whole genome sequencing data are protected and are not available due to data privacy laws. Source data are provided with this paper.

### Code availability

All code used for the analysis of data are provided in the Source Data file.

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

## Acknowledgements

The authors are extraordinarily grateful to all the patients who agreed to participate in this study. Without their participation, none of this could have happened. We also acknowledge the help and support from the Center for Biological Imaging at the University of Pittsburgh. Library quantification and Sequencing Library generation and sequencing were performed by the University of Pittsburgh Health Sciences Sequencing Core (HSSC), Rangos Research Center, UPMC Children's Hospital of Pittsburgh. We appreciate the support from Dr. Mary Armanios and the Molecular Diagnostics Lab at Johns Hopkins for providing telomere length measurements. We are indebted to Dr. Carol Greider for her critical feedback on this manuscript. This work was supported by NIH grants R01HL135062 (J.K.A.), R35ES030396, and R01CA207342 (P.L.O.), F31HL158063 (A.M.H.), and the UPMC Hillman Cancer Center Postdoctoral Fellowship for Innovative Cancer Research (S.L.S.). This project was supported in part by UPMC Hillman Cancer Center award P30CA047904.

## Author contributions

J.K.A. and A.M.H. conceived the study and overarching experimental design. S.L.S. and P.L.O. designed and analyzed biochemical analysis. R.M.S. analyzed the genetic data and family history. D.I.S., M.R.M., and J.F.M. provided clinical expertise. A.M.H., S.L.S., A.G.S., P.C., A.H.P., and K.E.L. performed experiments and analyzed data. J.K.A. and A.M.H. wrote the manuscript. All authors discussed the results and reviewed the manuscript.

## Competing interests

The authors declare no competing interests.
