## [Peer Review File · Nature Communications]

A persistent variant telomere sequence in a human pedigreeEditorial Note: This manuscript has been previously reviewed at another journal that is not operating a transparent peer review scheme. This document only contains reviewer comments and rebuttal letters for versions considered at *Nature Communications*.

REVIEWERS' COMMENTS

Reviewer #1 (Remarks to the Author):

As indicated in my previous review, I think that in the current form this paper presents a solid study that presents a very interesting and relevant finding. My concerns have been addressed experimentally and with appropriate text changes.

Reviewer #2 (Remarks to the Author):

In this revised manuscript, the authors have made considerable efforts to address several issues and have appropriately toned down statements that lacked full substantiation by the data. Consequently, the manuscript has undergone significant improvement.

However, I remain unconvinced by the experiment conducted with the POT1-delta OB construct (quantified in Figure 6, with primary data provided in supplemental figures).

The authors claim that "C50/56A TR was moderately enhanced by the addition of POT1OB, but not nearly to the extent of WT." According to them, these results suggest that the incorporation of a variant sequence would impede the binding of POT1, thereby promoting highly extendable telomeres through telomerase activity.

Despite the revisions, I find the data and its interpretation unconvincing. The authors place significant emphasis on this finding, as evidenced by the abstract's closing statement: "Incorporation of a variant sequence prevented POT1-mediated inhibition of telomerase and promoted telomere lengthening, partially offsetting the defect in enzyme processivity. These findings demonstrate that the shelterin component POT1 mediates an autoregulatory mechanism for maintaining the terminal telomere sequence by controlling access to the telomere end."

It would be beneficial for the authors to provide further clarification or additional experimental evidence to substantiate these claims. Alternatively, they should substantially temper this conclusion to align more closely with the available data.

Response to Reviewers

We understand that reviewing manuscripts is time consuming and we appreciate the time the reviews have spent on our behalf. We have addressed the concerns that were raised pointwise below.

Reviewer #1

As indicated in my previous review, I think that in the current form this paper presents a solid study that presents a very interesting and relevant finding. My concerns have been addressed experimentally and with appropriate text changes.

We appreciate the reviews suggestions to improve our manuscript and their appraisal of our revised manuscript.

Reviewer #2

The authors claim that "C50/56A TR was moderately enhanced by the addition of POT1OB, but not nearly to the extent of WT." According to them, these results suggest that the incorporation of a variant sequence would impede the binding of POT1, thereby promoting highly extendable telomeres through telomerase activity.

Despite the revisions, I find the data and its interpretation unconvincing. The authors place significant emphasis on this finding, as evidenced by the abstract's closing statement: "Incorporation of a variant sequence prevented POT1-mediated inhibition of telomerase and promoted telomere lengthening, partially offsetting the defect in enzyme processivity. These findings demonstrate that the shelterin component POT1 mediates an autoregulatory mechanism for maintaining the terminal telomere sequence by controlling access to the telomere end."

It would be beneficial for the authors to provide further clarification or additional experimental evidence to substantiate these claims. Alternatively, they should substantially temper this conclusion to align more closely with the available data.

We have re-written our abstract to de-emphasize this finding as requested by the reviewer. The abstract now reads, "Incorporation of a variant sequence prevented POT1-mediated inhibition of telomerase suggesting that incorporation of a variant sequence may influence telomere addition. These findings demonstrate that telomeres can tolerate substantial degeneracy while remaining functional and provide insights as to how incorporation of a non-canonical telomere sequence might alter telomere length dynamics." We also adjusted the wording in the results and discussion to soften our conclusions around this point. We have also added additional discussion that highlights that this phenomenon has been reported previously and softened our conclusions in the discussion. "Previous studies in the budding yeast *Kluyveromyces lactis* found that specific mutation that alter the telomeric DNA sequence caused dramatic telomere lengthening⁴⁹ and mammalian cells that co-overexpress wild type and variant TR have longer telomeres than those that over express wild type TR alone²⁹. Taken together, these data suggest that when variant telomere sequences are incorporated, they can have

conflicting effects on telomerase activity and telomere length regulation.” We hope that the reviewer feels that this is a reasonable and measured interpretation of the data.